# Hematopoietic stem and progenitor cells improve survival from sepsis by boosting immunomodulatory cells

Daniel E Morales-Mantilla[1,2], Bailee Kain[2,3], Duy Le[1,2], Anthony R Flores[4], Silke Paust[5], Katherine Y King[1,2,3]*

[1]Graduate Program in Immunology, Baylor College of Medicine, Houston, United States; [2]Department of Pediatrics, Division of Infectious Diseases, Baylor College of Medicine, Houston, United States; [3]Graduate Program in Translational Biology and Molecular Medicine, Baylor College of Medicine, Houston, United States; [4]Division of Infectious Diseases, Department of Pediatrics, UTHSC/McGovern Medical School, Houston, United States; [5]The Scripps Research Institute, Department of Immunology and Microbiology, La Jolla, United States

*For correspondence:
kyk@bcm.edu

Competing interest: The authors declare that no competing interests exist.

## Abstract

New therapeutic strategies to reduce sepsis-related mortality are urgently needed, as sepsis accounts for one in five deaths worldwide. Since hematopoietic stem and progenitor cells (HSPCs) are responsible for producing blood and immune cells, including in response to immunological stress, we explored their potential for treating sepsis. In a mouse model of Group A *Streptococcus* (GAS)-induced sepsis, severe immunological stress was associated with significant depletion of bone marrow HSPCs and mortality within approximately 5–7 days. We hypothesized that the inflammatory environment of GAS infection drives rapid HSPC differentiation and depletion that can be rescued by infusion of donor HSPCs. Indeed, infusion of 10,000 naïve HSPCs into GAS-infected mice resulted in rapid myelopoiesis and a 50–60% increase in overall survival. Surprisingly, mice receiving donor HSPCs displayed a similar pathogen load compared to untreated mice. Flow cytometric analysis revealed a significantly increased number of myeloid-derived suppressor cells in HSPC-infused mice, which correlated with reduced inflammatory cytokine levels and restored HSPC levels. These findings suggest that HSPCs play an essential immunomodulatory role that may translate into new therapeutic strategies for sepsis.

## Editor's evaluation

This preclinical study reports on a novel strategy for sepsis. Sepsis induced by Group A Streptococcus (GAS) in mice leads to depletion of bone marrow HSPCs and mortality and infusion of naive donor HSPCs lower mortality but has no effect on bacterial burden. This supports that HSPCs infusion might attenuate the detrimental immune response in sepsis warranting further investigation of this novel concept.

## Introduction

Sepsis accounts for one in five deaths worldwide and is a common final pathway for many disease processes such as cancer, diabetes, and cardiovascular disease (*Rhee et al., 2019*). Sepsis is an inflammatory syndrome largely driven by the activation of immune cells by pathogen associated molecular patterns (PAMPs) and damage-associated molecular patterns (DAMPs) (*van der Poll et al., 2017*; *Deutschman and Tracey, 2014*). After recognizing these molecules via pattern recognition receptors,

immune cells become activated and produce proinflammatory cytokines, notably interleukins IL-1, IL-6, interferons (IFNs), and tumor necrosis factor (TNF) that contribute to fever, vasodilation, and multiorgan dysfunction (*Deutschman and Tracey, 2014*; *Yu et al., 2011*; *Huang et al., 2005*; *Wang et al., 2008*). For patients that progress to septic shock, mortality rates remain as high as 40% (*Napolitano, 2018*).

Leukopenia is a feature of severe sepsis that arises from apoptosis of peripheral immune cells and is an independent risk factor for death. To counteract the adverse effects of leukopenia, investigators have used immunotherapies such as GM-CSF (*Krebs et al., 2019*) or granulocyte infusions in an attempt to restore leukocyte numbers and improve survival. These strategies have produced mixed results (*Estcourt et al., 2016*; *Li et al., 2019*; *Mathias et al., 2015*). The benefits of granulocyte infusions in cancer patients with fever and neutropenia are limited by the difficulty of obtaining sufficient cells and the short-lived nature of those cells (*Hidalgo et al., 2019*; *Robinson and Marks, 2004*).

Recent work from our group and others indicates that hematopoietic stem and progenitor cells (HSPCs) express surface receptors for cytokines, chemokines, and PAMPs (*Karpova et al., 2017*; *Burberry et al., 2014*; *Nagai et al., 2006*; *Schürch et al., 2014*; *Matatall et al., 2014*; *Baldridge et al., 2011*; *Pietras et al., 2016*; *Takizawa et al., 2017*) and respond rapidly upon direct and indirect stimulation by these signals. HSPCs, the progenitors of all blood and immune cells, are comprised of five subgroups of hematopoietic cells: hematopoietic stem cells (HSCs), which have long-term self-renewal capacity, and four types of multipotent progenitors (MPPs 1–4), which are defined by lower self-renewal capacity and myeloid or lymphoid differentiation biases (*Wang et al., 2008*; *Morales-Mantilla and King, 2018*; *Chambers et al., 2007*; *Sun et al., 2014*; *Rodriguez-Fraticelli et al., 2018*; *Cabezas-Wallscheid et al., 2014*). Immune responses induce HSPCs in the bone marrow (BM) to produce effector immune cells via a process called emergency hematopoiesis (*Matatall et al., 2014*; *Pietras et al., 2016*; *Takizawa et al., 2017*; *Morales-Mantilla and King, 2018*; *MacNamara et al., 2011*; *Matatall et al., 2016*). The capacity of HSPCs to directly detect pathogen-derived molecules, cytokines, and chemokines suggests that emergency granulopoiesis can be mobilized from even the most primitive hematopoietic progenitors and that HSPCs have an active role in fighting infections. However, the extent and mechanism by which HSPC responses contribute to immunity in the acute setting remain poorly defined.

We recently showed that chronic inflammatory stress impairs HSPC quiescence and self-renewal while promoting their activation and terminal differentiation (*Matatall et al., 2016*). Upon direct sensing of inflammatory cytokines such as interferon-gamma (IFNγ), HSCs are dislodged from their normal position near quiescence-enforcing CXCL12-abundant reticular cells in the niche. Inflammatory signaling induces transcription factors such as Pu.1, CEBPb, and BATF2 (*Matatall et al., 2014*; *Pietras et al., 2016*; *Matatall et al., 2016*; *Sato et al., 2020*) to promote myeloid differentiation, leading to the expansion of granulocyte and monocyte populations. Disruption of the homeostatic balance of self-renewal and differentiation eventually leads to depletion of the progenitor compartment (*Pietras et al., 2016*; *Morales-Mantilla and King, 2018*; *Matatall et al., 2016*). Collectively, these studies point toward a direct role for HSPCs in supplying the myeloid cells critical to the immune response against infection.

To test their contribution to immune responses during acute infection, we examined the role of HSPCs in a mouse model of *Streptococcus pyogenes* infection, also known as Group A *Streptococcus* (GAS). GAS is a common and clinically relevant pathogen that causes a plethora of diseases, from mild skin infections to life-threatening necrotizing fasciitis and sepsis (*Wang et al., 2008*; *Efstratiou and Lamagni, 2016*; *Emgård et al., 2019*; *Walker et al., 2014*). GAS infections can infiltrate the bloodstream and other organs, causing high systemic levels of inflammatory cytokines including IFNγ, TNF, IL-1, and IL-6. As HSPCs have been shown to activate and differentiate in response to these cytokines (*Burberry et al., 2014*; *Matatall et al., 2014*; *Pietras et al., 2016*; *Takizawa et al., 2017*; *Morales-Mantilla and King, 2018*; *Gong et al., 2020*; *Esplin et al., 2011*; *Chou et al., 2012*), in this study we sought to determine the role of HSPCs in immune responses against infections.

Here, we found that GAS infection significantly depletes HSPCs in the bone marrow (BM). We tested the idea of infusing HSPCs to restore the hematopoietic progenitor pool. Mice treated with HSPCs showed restored HSPC numbers in the BM, increased myeloid cell production, and significantly improved overall survival. Surprisingly, HSPC infusion did not reduce pathogen burden. Instead, HSPC infusion correlated with a significant increase in the abundance of myeloid-derived suppressor

cells (MDSCs) and a dampening of overall systemic inflammation. In summary, our studies indicate that HSPCs contribute to survival from sepsis by supporting the production of immunosuppressive MDSCs.

## Results

### GAS infection induces trafficking of myeloid cells from the BM into circulation

To characterize the impact of acute infections on the hematopoietic system, we inoculated mice by intramuscular injection of the hind leg with $2 \times 10^6$ colony forming units (CFU) of the model pathogen *S. pyogenes* strain MGAS315. To characterize differentiated hematopoietic populations during infection, we performed flow cytometric analysis of BM and peripheral blood (PB) lineage cells 24 hr after GAS infection (*Figure 1A*) and collected serum for cytokine analyses. BM characterization of lineage cells showed a significant decrease in BM monocytes (*Figure 1C*) and granulocytes (*Figure 1D*) with no change in BM B or T cells (*Figure 1E and F*). In contrast, PB lineage composition was significantly skewed toward myeloid cells with significantly higher circulating monocytes and granulocytes (*Figure 1G and H*) and lower lymphoid cells (*Figure 1I and J*). Serum cytokine characterization showed a significant increase in monocyte chemoattractant protein-1 (MCP-1; also known as CCL2) (*Figure 1B*). These results suggest that myeloid cells exit the BM into circulation following an MCP-1 gradient, consistent with prior studies showing MCP-1-driven mobilization during inflammation (*Tsou et al., 2007*; *Schultze et al., 2019*).

### GAS infection depletes bone marrow HSPCs without evidence of extramedullary hematopoiesis

After 24 hr of infection, the state of HSPCs in the infected mice were analyzed by flow cytometry of BM and spleen (*Figure 2A*) (see *Table 1* for surface markers). BM cells were not gated for the common stem cell marker SCA1 (*Figure 2B*), since it has been previously described to be non-specifically expressed during inflammatory stress (*Baldridge et al., 2011*). The total number of HSPCs dropped significantly in just 24 hrs in GAS-infected mice (*Figure 2C*). More specifically, HSPC subpopulations including HSCs, multipotent progenitor 3 (MPP3s), and MPP4s were significantly lower in GAS-infected mice (*Figure 2D, E, and F*).

Extramedullary hematopoiesis is the proliferation and differentiation of HSCs in tissues other than the BM, the canonical stem cell niche. The spleen is one of the most common sites of extramedullary hematopoiesis during infections (*Yang et al., 2020*). To assess whether a reciprocal increase in extramedullary hematopoiesis accompanied the loss of HSPCs in the BM, we analyzed spleen tissue by flow cytometry. While there was a slight increase in total HSPCs in the spleen (*Figure 2G*), there was no significant change in spleen populations that include HSCs/MPP1, MPP2s, or MPP3/4 (*Figure 2H-J*). These findings suggest that the loss of BM HSPC populations is not principally a result of migration from the BM into the spleen, and other mechanisms such as terminal differentiation (*Matatall et al., 2016*) also likely contribute to the noted HSPC depletion, as observed in other studies.

### GAS infection induces HSC myeloid differentiation

Activation of HSPCs by PAMPs or cytokines promotes their proliferation and differentiation (*Nagai et al., 2006*; *Matatall et al., 2014*; *Pietras et al., 2016*; *Takizawa et al., 2017*; *Morales-Mantilla and King, 2018*; *Baldridge et al., 2010*). To determine the lineage fate of endogenous HSPCs following GAS infection, we performed lineage tracing experiments using the tamoxifen-inducible KRT18-CreERT2: Rosa26-lox-STOP-lox-TdTomato mouse system (*Figure 3A*). Within the BM, *Krt18* is almost exclusively expressed in HSCs (*Chapple et al., 2018*) and these mice do not have any immunological impairment that would change the severity of our infection model. Tamoxifen induction activates the CreERT2 protein in Krt18-expressing HSCs, resulting in irreversible TdTomato expression in HSCs and their newly formed progeny (*Figure 3B*).

After 5 days of intraperitoneal injections of tamoxifen, mice were inoculated with GAS or saline. Since the average mammalian cell cycle takes 24 hr, we decided to trace the lineage of hematopoiesis 72 hr post GAS infection. After these 72 hr, BM and PB was harvested for flow cytometric analysis. Analysis of the BM showed that GAS infection induced the production of new HSPCs, which includes short-term HSCs and MPPs (*Figure 3C-D*). In addition, there was significant labeling of CD41+ HSCs,

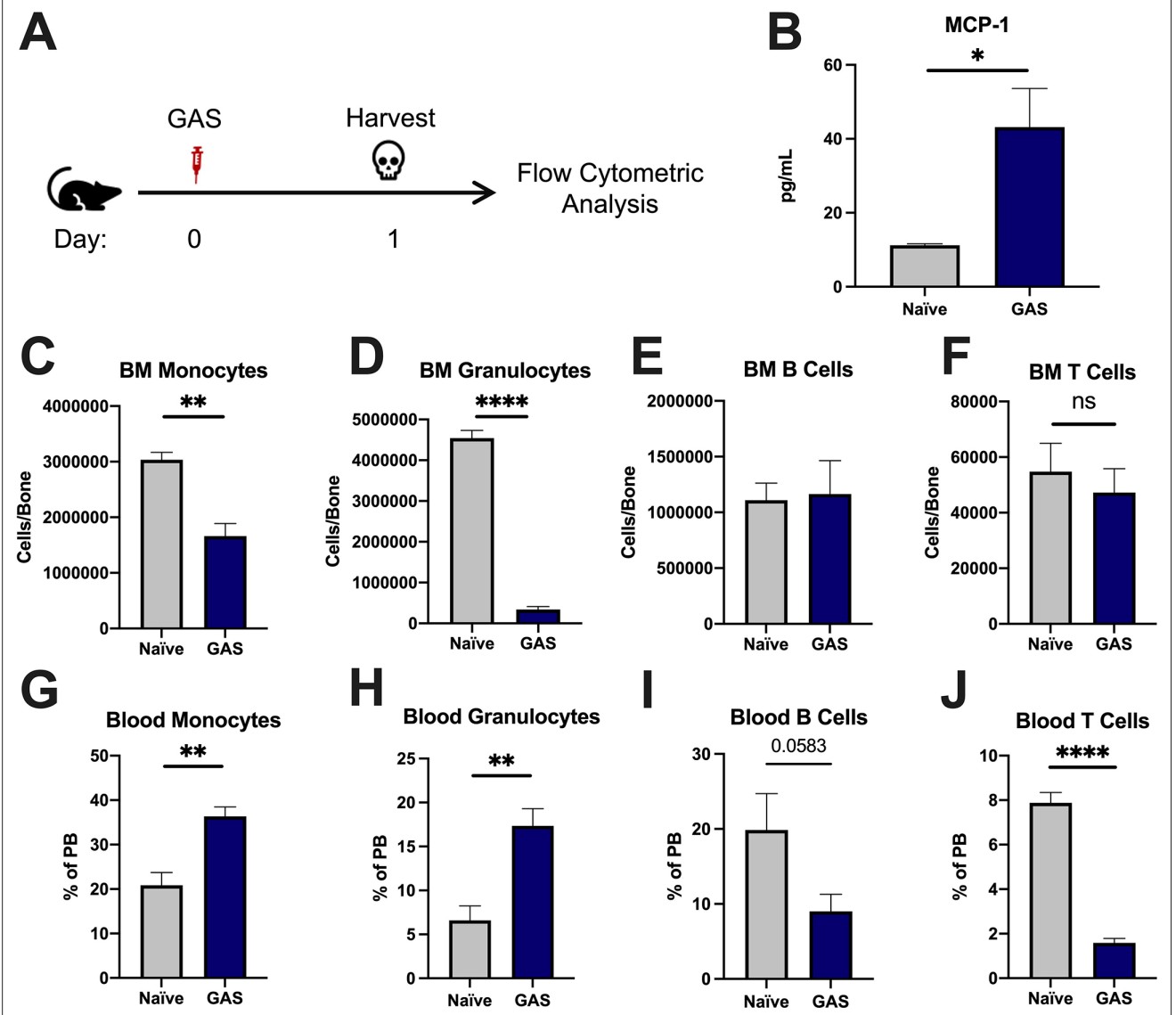

**Figure 1.** Group A *Streptococcus* (GAS) infection promotes a rapid myeloid cell response. (**A**) Experimental time frame of GAS infection and bone marrow (BM) analysis. (**B**) Serum levels of monocyte chemoattractant protein-1 (MCP-1) of naïve and GAS-infected mice. Absolute number of (**C**) monocytes, (**D**) granulocytes, (**E**) B cells, and (**F**) F cells in the BM of naïve and infected mice. Relative abundance of (**G**) monocytes, (**H**) granulocytes, (**I**) B cells, and (**J**) T cells in the blood. (**B–J**) Data is representative of three independent experiments; n = 3–5 mice/group; statistical comparison done using unpaired t-test; *p < 0.05, **p < 0.01, ****p < 0.0001. Outliers were identified using the ROUT method (Q = 5%).

The online version of this article includes the following source data and figure supplement(s) for figure 1:

**Source data 1.** BM and PB populations of lineage cells and serum MCP-1 levels.

**Figure supplement 1.** Group A *Streptococcus* (GAS) infection and superinfection promotes exit of myeloid cells from bone marrow (BM) into circulation.

**Figure supplement 1—source data 1.** BM counts of HSPC populations of naïve, infected, and superinfected mice.

a myeloid-biased and proinflammatory subset of HSCs (**Figure 3E**; **Gekas and Graf, 2013**), and new cells of the myeloid lineage (**Figure 3F-H**). While we found a significant increase in new BM monocytes (**Figure 3G**), there was no significant change in the frequency of TdTomato-labeled granulocyte/ monocyte progenitors (GMPs) (**Figure 3I**), which may simply reflect a rapid flow through this compartment to terminally differentiated populations. We also saw no statistically significant increase in BM granulocytes (**Figure 3H**); however, PB analysis showed a significant increase in new myeloid cells in both monocytic and granulocytic branches (**Figure 3L-N**). While there was a significant decrease in

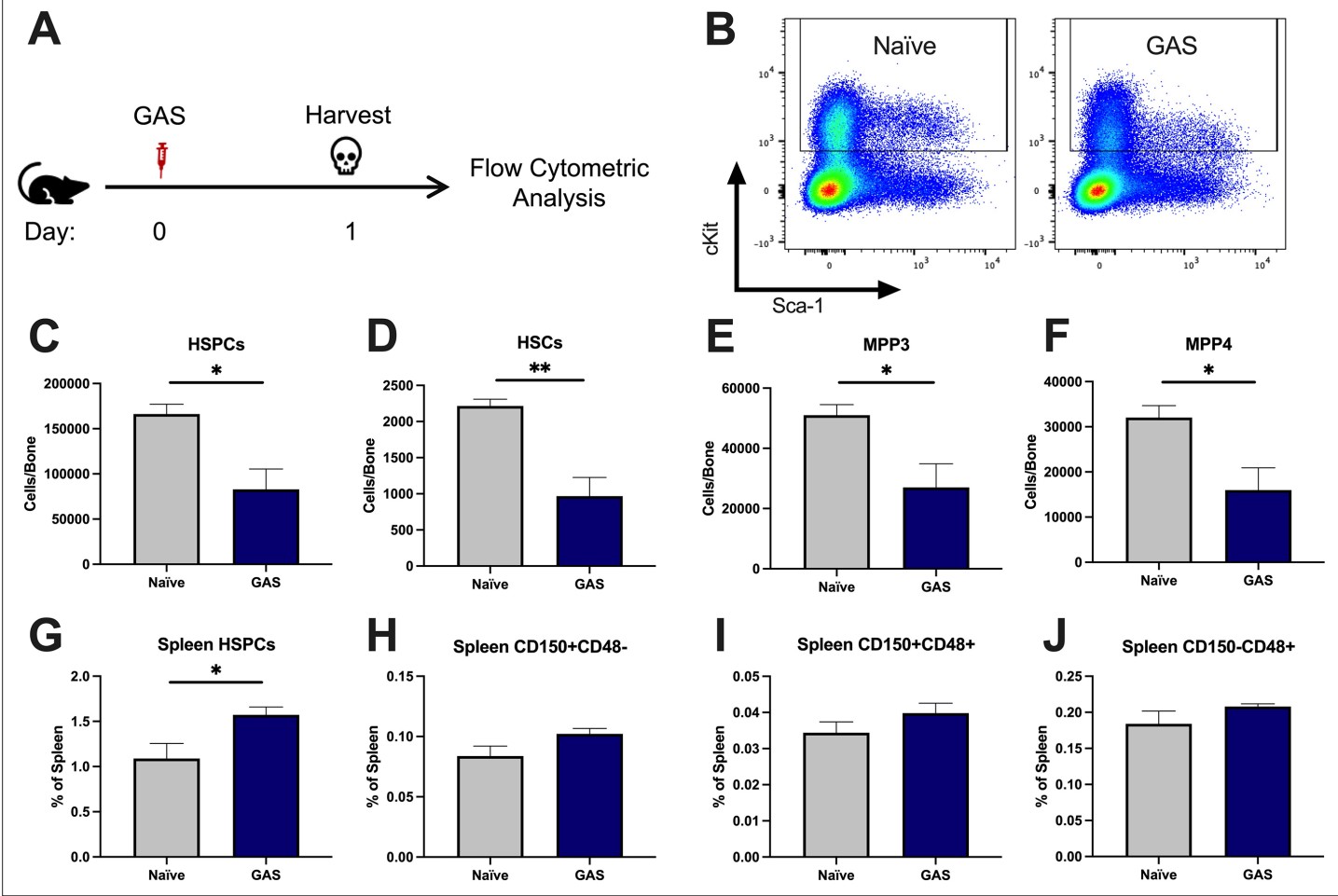

**Figure 2.** Group A *Streptococcus* (GAS) infection depletes bone marrow (BM) hematopoietic stem and progenitor cells (HSPCs) 24 hr post infection. (**A**) Experimental time frame of GAS infection and BM analysis. (**B**) Flow plot of HSPC gating and representation of different surface expression of cKit and Sca-1 during infection. Plots are gated from lineage negative BM cells. (**C–F**) Absolute number of HSPCs, hematopoietic stem cells (HSCs), multipotent progenitors (MPP) 3, and MPP4 in the BM of naïve and GAS-infected mice. Spleen populations of (**G**) HSPCs, (**H**) HSC/MPP1, (**I**) MPP2, and (**J**) MMP3/4 identified by differential expression of CD150 and CD48. (**C–J**) Data is representative of three independent experiments; n = 4–5 mice/group; statistical comparison done using unpaired t-test; *p < 0.05, **p < 0.01. Outliers were identified using the ROUT method (Q = 5%).

The online version of this article includes the following source data and figure supplement(s) for figure 2:

**Source data 1.** BM counts of HSPC populations of naïve and infected mice and spleen HSPC populations.

**Figure supplement 1.** Superinfection further depletes hematopoietic stem and progenitor cells (HSPCs) and hematopoietic stem cells (HSCs) in the bone marrow (BM).

**Figure supplement 1—source data 1.** BM and PB levels of immune cells and MCP-1 during infection.

---

the production of new BM T cells (***Figure 3J***), there was no change in BM B cells (***Figure 3K***) nor PB T or B cells (***Figure 3O–P***). These data suggest that endogenous HSCs undergo rapid emergency myelopoiesis during GAS infection.

## HSPC infusion promotes survival in GAS-infected mice

Given that HSPCs are activated to divide and differentiate into immune effector cells upon inflammatory stimulation and we observed an acute loss of HSPCs in GAS-infected mice, we hypothesized that infusion of naïve HSPCs (Lin- Sca-1+ c-Kit+) into GAS-infected mice could improve pathogen clearance, reduce tissue damage, and prolong survival. To test this hypothesis, we infected mice with $2 \times 10^6$ CFU MGAS315 and then infused 10,000 FACS-purified HSPCs 24 hr later, when endogenous HSPCs are significantly decreased (***Figure 4A***). This HSPC dose, equivalent to approximately $1.7 \times 10^7$ cells per m$^2$ body surface area, is significantly lower than the dose used for granulocyte

**Table 1.** Surface markers for flow cytometry. Hematopoietic cell populations identified by flow cytometry were characterized using the listed surface markers. Lineage (Lin) markers include Gr1, CD11b, B220, CD4, CD8, and Ter119.

| Population | Markers |
| --- | --- |
| HSC | Lin- ckit+ CD150+ CD48- CD34- Flk2- |
| CD41+ HSCs | Lin- ckit+ CD150+ CD48- CD34- Flk2- CD41+ |
| Donor HSPCs | Lin- ckit+ Sca1+ |
| HSPCs | Lin- ckit+ |
| MPP1 | Lin- ckit+ CD150+ CD48 CD34+ Flk2- |
| MPP2 | Lin- ckit+ CD150+ CD48+ Flk2- |
| MPP3 | Lin- ckit+ CD150 CD48+ CD34+ Flk2- |
| MPP4 | Lin- ckit+ CD150 CD48+ CD34+ Flk2+ |
| GMP | Lin- ckit+ CD41- CD150- CD16/32+ |
| MkP | Lin- ckit+ CD150+ CD41+ |
| Myeloid cells | Gr1+ Mac1+ B220- CD4- CD8- |
| B cells | Gr1- Mac1- B220+ CD4- CD8- |
| T cells | Gr1- Mac1- B220- CD4+ CD8 or Gr1- Mac1- B220- CD4- CD8+ |
| Granulocytes | Gr1+ Mac1+ B220- CD4- CD8- SSC-mid F4/80- |
| Eosinophils | Gr1- Mac1+ B220- CD4- CD8- SSC$^{Hi}$ |
| Macrophages | Gr1+ Mac1+ B220- CD4- CD8- SSC$^{low}$ F4/80+ |
| Monocytes | Mac1+ B220- CD4- CD8- SSC$^{low}$ |
| PMN-MDSC | CD11b + Ly6G + Ly6C$^{low}$CD244+ |
| M-MDSC | CD11b + Ly6G- Ly6C$^{hi}$ |

infusions, typically between $10^8$ and $10^{10}$ cells per m$^2$ (*Price et al., 2015*). On day 3 post infection, we harvested BM, limb tissue, and spleen to characterize BM populations and pathogen load (*Figure 4B*).

BM characterization showed that HSPC infusion restored the relative abundance of HSPCs, HSCs, and myeloid-biased progenitors, such as MPP3s and GMPs, in the BM (*Figure 4C-H*). We observed that GAS-infected mice that received HSPCs showed lower morbidity than non-rescued mice, with improved overall body score and activity level. To assess survival, we performed Kaplan-Meier survival studies of GAS-infected mice in the absence or presence of HSPC rescue (*Figure 4K*). GAS-infected mice infused with HSPCs had significantly higher survival than non-rescued mice (*Figure 4L*). However, to our surprise, HSPC infusion did not significantly affect the pathogen burden in the infected muscle (*Figure 4I*) or pathogen spread to other tissues (*Figure 4J*). Overall, these findings suggest that HSPC infusion is beneficial during GAS infections and promotes survival by a mechanism other than pathogen clearance.

## Superinfection further depletes HSPCs in mice

To determine the extent of the protective potential of HSPC infusion during infections, we tested the efficacy of HSPC rescue in a mouse model of influenza and GAS superinfection. Here, we used a model of influenza and *S. pyogenes* (strain MGAS315) bacterial superinfection. Mice were infected with influenza (strain H1N1 PR8) by intranasal injection of 150 plaque-forming units (PFU). On day 3 post influenza virus infection, which represents peak viral replication for humans and mice (*Baccam et al., 2006*; *Smith and Perelson, 2011*), we injected mice with 2 × $10^6$ CFU MGAS315 by IM inoculation (*Figure 2—figure supplement 1A*). On day 4 (24 hr post GAS infection), we analyzed BM and PB. Lin- cells in the BM showed phenotypical differences in surface expression of Sca-1 and c-Kit proteins depending on the pathogen combination (*Baldridge et al., 2011*; *Figure 2—figure supplement 1B*). In addition, superinfection caused a severe decrease in HSPCs and their subpopulations (*Figure 2—figure supplement 1C–1H*). Most notably, the absolute number of HSCs was reduced to 20–30% of a healthy mouse (*Figure 2—figure supplement 1D*).

We also analyzed BM and PB lineage populations of superinfected mice (*Figure 1—figure supplement 1A*). Similar to GAS-infected mice, superinfection led to an increase in serum levels of MCP-1 (*Figure 1—figure supplement 1B*) that resulted in the exit of BM monocytes and BM granulocytes (*Figure 1—figure supplement 1C, G*) into circulation (*Figure 1—figure supplement 1D, H*). BM B cell and T cell numbers did not change (*Figure 1—figure supplement 1E, I*), while the abundance of circulating B and T cells was reduced in GAS-infected and superinfected mice (*Figure 1—figure supplement 1F, J*).

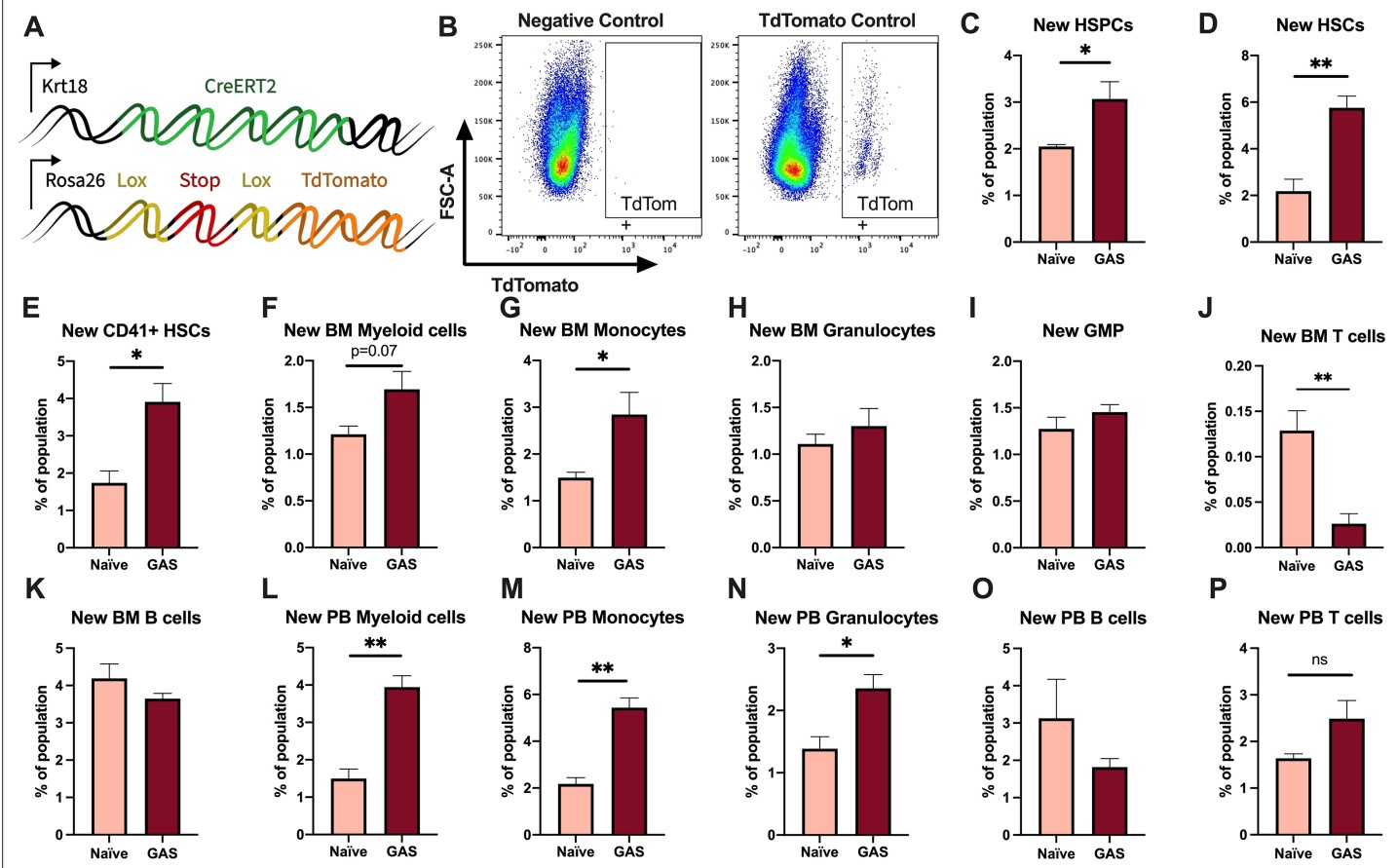

**Figure 3.** Krt18 lineage tracing of naïve and Group A *Streptococcus* (GAS)-infected hematopoietic stem cells (HSCs). (**A**) Genetic model of KRT18-CreERT2:Rosa26-lox-STOP-lox-TdTomato mouse system. (**B**) Representative gating of TdTomato expression in negative control (left: Genotype: Krt18-CreERT2+) and tamoxifen-induced positive control (right: Genotype: KRT18-CreERT2+: Rosa26-lox-STOP-lox-TdTomato+/-). Percent of bone marrow (BM) (**C**) hematopoietic stem and progenitor cell (HSPC), (**D**) HSC, (**E**) CD41+ HSC, and (**F**) myeloid populations that are TdTomato+. Percent of BM (**G**) total monocytes, (**H**) granulocytes, (**I**) granulocyte/monocyte progenitor (GMP), (**J**) BM T cells, and (**K**) BM B cells that are TdTomato+. Percent of peripheral blood (PB) (**L**) total myeloid cells, (**M**) monocytes, (**N**) granulocytes, (**O**) T cells, and (**P**) B cells that are TdTomato+. Data representative of three independent experiments; (**C–N**) n = 5–7 mice/group. Statistical comparison done using unpaired t-tests; *p < 0.05, **p < 0.01.

The online version of this article includes the following source data for figure 3:

**Source data 1.** Levels of newly generated cells in response to GAS infection.

## HSPC infusion promotes survival in superinfected mice and increases levels of HSPCs and myeloid progenitors in the BM

The loss of HSPCs and HSCs was very prominent in superinfected mice, more so than in mice infected with GAS alone. Therefore, we hypothesized that an infusion of 10,000 HSPCs would also benefit mice in this model of superinfection (*Figure 5A*). As expected, HSPC infusion significantly increased HSPCs and myeloid-biased progenitors in superinfected mice (*Figure 5B-G*). As seen in GAS-infected mice, HSPC infusion did not promote bacterial (*Figure 5H*) or viral (*Figure 5I*) clearance in superinfected mice. The spread of bacteria to the spleen was also unaffected by HSPC infusion (*Figure 5J*).

Despite the severity of the infection, superinfected mice that received an HSPC infusion (*Figure 5K*) had significantly improved survival compared to non-rescued mice (*Figure 5L*). This finding suggests that the protective properties of HSPC infusion are effective even in this very severe model of infection.

## HSPC infusion increases immunomodulatory MDSCs and prevents sepsis-induced cytokine exacerbation

Production of proinflammatory cues including IL1, IL6, IL8, TNF, and MIP1a is a key driver of morbidity during sepsis. Together, these cues contribute to systemic inflammatory response syndrome (SIRS),

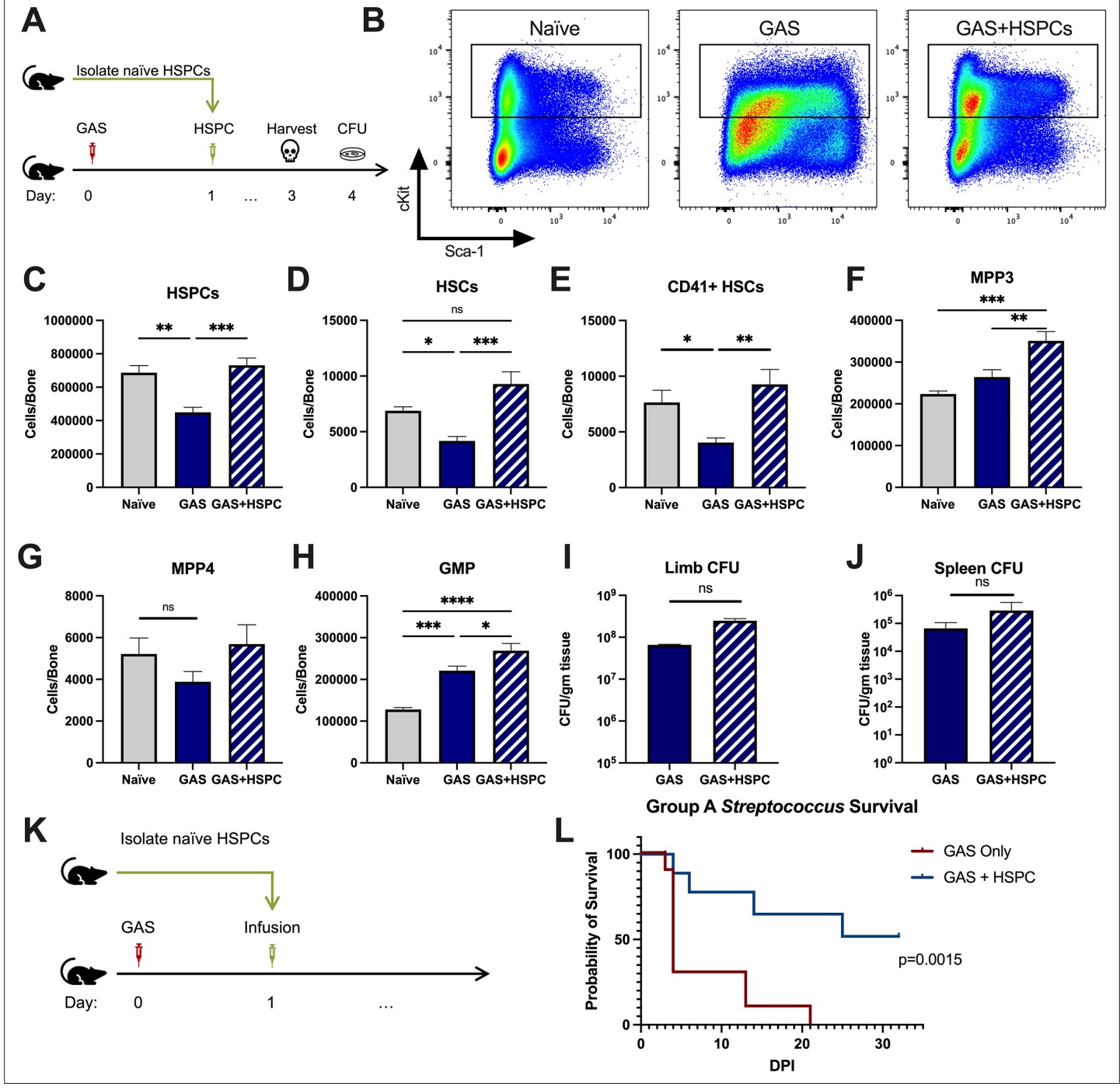

**Figure 4.** Hematopoietic stem and progenitor cell (HSPC) infusion in Group A *Streptococcus* (GAS)-infected mice promotes survival and increases progenitor populations in the bone marrow (BM). (**A**) Experimental design of BM analysis and colony forming unit (CFU) count after HSPC infusion. (**B**) Flow plot of HSPC gating and representation of different surface expression of cKit and Sca-1 during infection. Plots are gated from lineage negative BM cells. (**C–H**) Absolute numbers of HSPCs, hematopoietic stem cells (HSCs), and downstream progenitors in the BM of naïve, GAS-infected mice, and GAS-infected mice rescued with HSPCs. Quantified bacterial load in the (**I**) limb and (**J**) spleen of infected mice. (**K**) Experimental design of the survival study. (**J**) Survival curve of GAS-infected mice with or without HSPC infusion. Data representative of three independent experiments; (**C–H**) n = 5–7 mice/group, (I and J) n = 8–10 mice/group, (**L**) n = 9–10 mice/group. Statistical comparison done using (**C–H**) one-way ANOVA with Tukey's correction for multiple comparisons or (I and J) unpaired t-tests. ns = not significant, *p < 0.05, **p < 0.01, ***p < 0.001, *****p < 0.0001. Comparison of (**L**) survival was done using Log-rank (Mantel-Cox) test.

The online version of this article includes the following source data for figure 4:

**Source data 1.** HSPC levels after infusion and survival data.

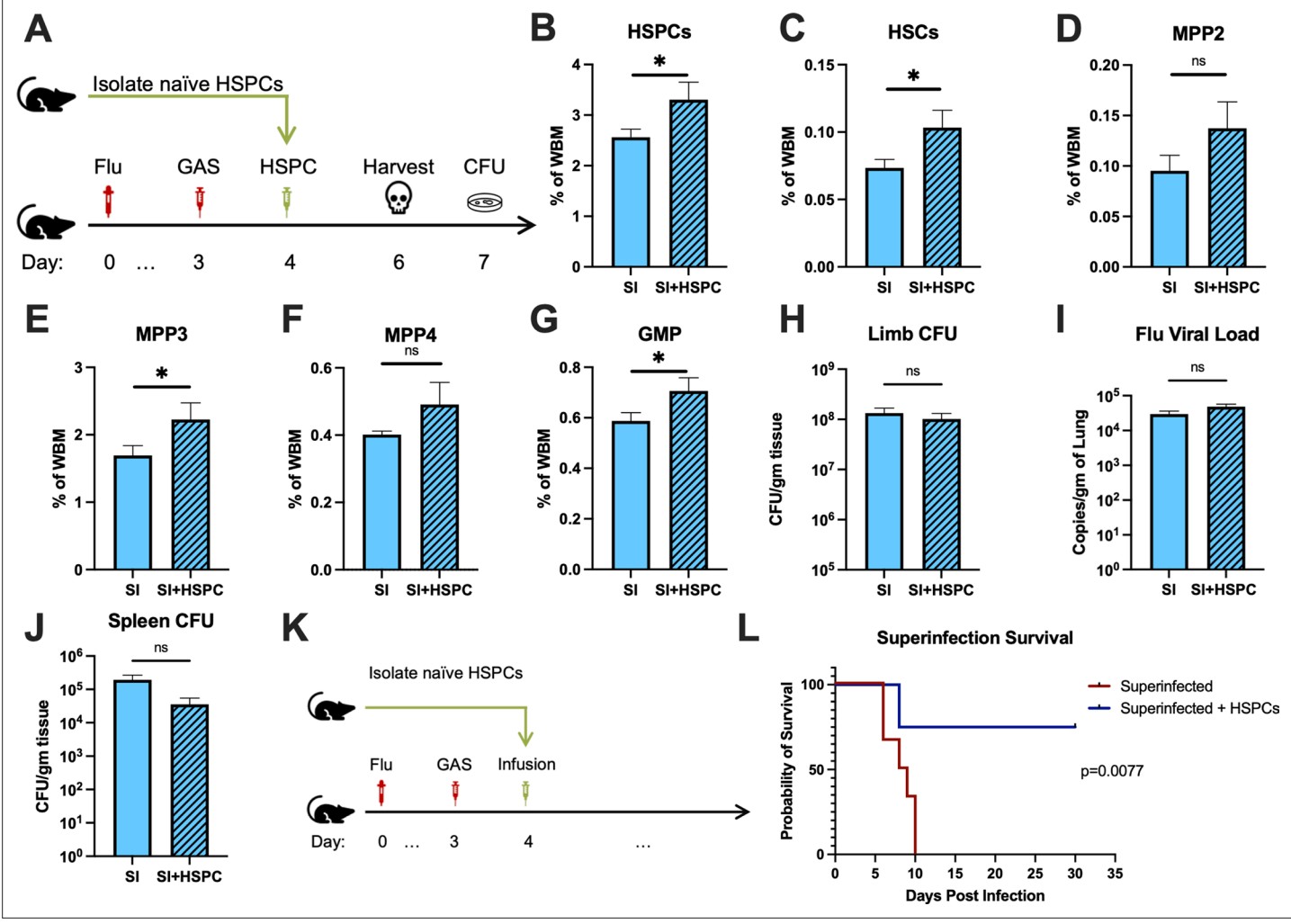

**Figure 5.** Hematopoietic stem and progenitor cell (HSPC) infusion in superinfected mice promotes survival without changing pathogen clearance. (**A**) Experimental design of bone marrow (BM) analysis and colony forming unit (CFU) count post HSPC infusion in superinfected mice. (**B–G**) BM populations of HSPCs and downstream progenitors after HSPC infusion. Bacterial load in the (**H**) limb and (**J**) spleen of infected mice. (**I**) Viral load of mice with or without HSPC infusion. (**K**) Experimental design of the survival studies on superinfected mice. (**L**) Survival curve after HSPC infusion. Experiments are representative of three independent experiments. (**B–J**) Comparison done with unpaired t-test or Welch's t-test. (**B–G**) n = 5, (**H–J**) n = 9–11, and (**L**) n = 9–10 mice per group. Comparison of (**L**) survival was done using Log-rank (Mantel-Cox) test. ns = not significant, *p < 0.05.

The online version of this article includes the following source data for figure 5:

**Source data 1.** HSPC levels in superinfected mice after infusion and survival data.

including fever, tachypnea, vasodilation, and circulatory collapse (*Huang et al., 2005*; *Jaffer et al., 2010*). These cytokines are independently associated with poor outcomes and death from sepsis in humans. Since we observed improved survival in mice receiving HSPC infusion without any changes in pathogen load, we hypothesized that HSPCs could impact immunomodulatory cell composition and the inflammatory response to severe infection. Upon analysis of PB and BM populations 3 days after infection and 2 days after HSPC infusion (*Figure 6A*), there were no changes in BM or PB T lymphocytes that could indicate a Treg-related activity (*Figure 6—figure supplement 1A* and 1B). However, we found that HSPC infusion significantly increased PB polymorphonuclear MDSCs (PMN-MDSCs) (*Figure 6B*) and restored PB monocytic-MDSCs (M-MDSCs) levels (*Figure 6C*) 3 days post infection and 2 days post HSPC infusion. Similarly, HSPC infusion restored BM PMN-MDSCs and M-MDSCs populations in GAS-infected mice (*Figure 6D and E*). These cells were functionally validated (*Figure 6—figure supplement 1C*) as immunosuppressive cells by their ability to reduce activated T cell proliferation in culture (*Figure 6—figure supplement 1D*). Strikingly, cytokine profiling

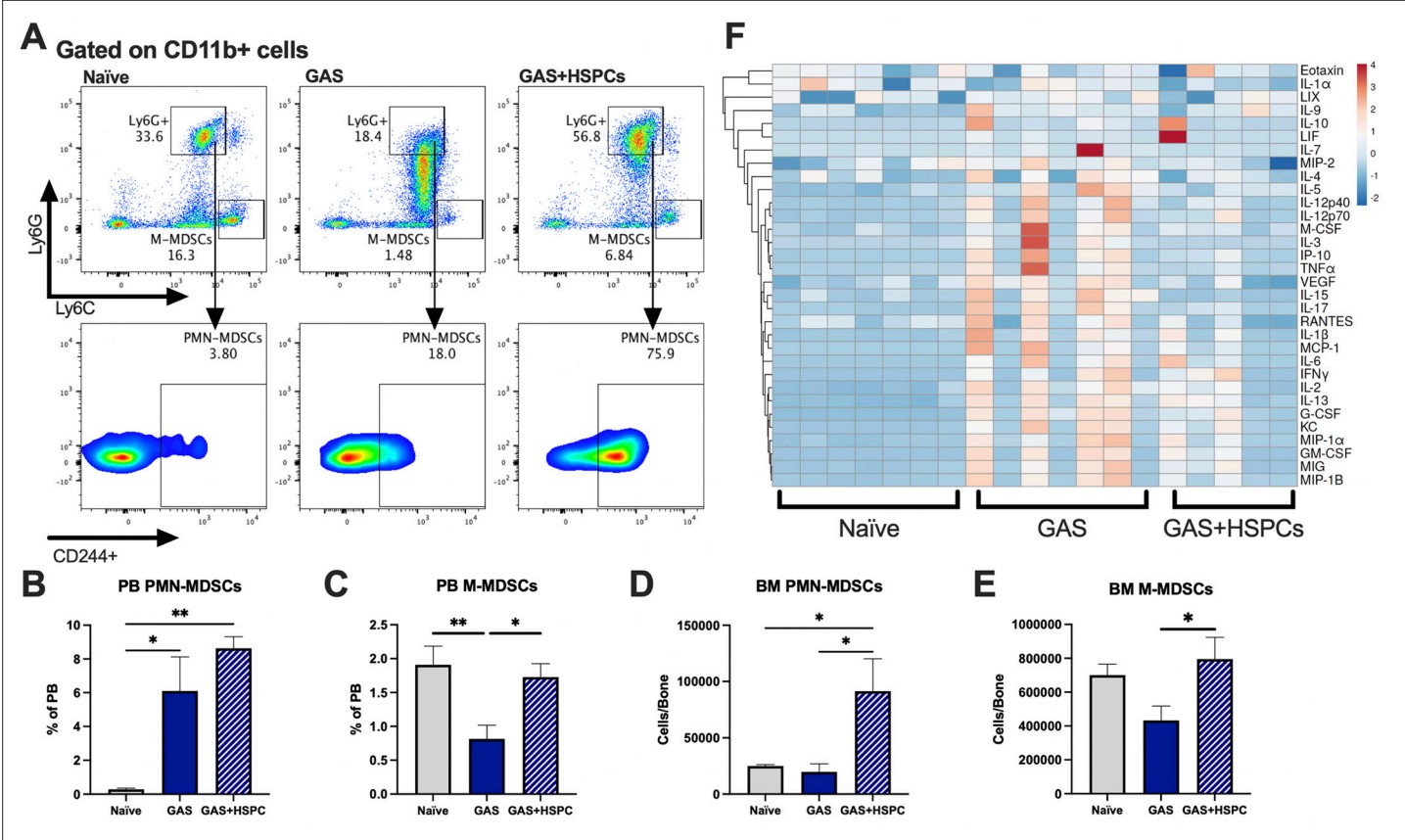

**Figure 6.** Hematopoietic stem and progenitor cell (HSPC) infusion increases and restores myeloid-derived suppressor cell (MDSC) populations and dampens inflammation after Group A *Streptococcus* (GAS). (**A**) Representative gating of MDSCs by their surface expression of Ly6G and Ly6C at day 3 post infection. Gated on CD11b + cells. Peripheral blood (PB) populations of (**B**) polymorphonuclear MDSCs (PMN-MDSCs) and (**C**) monocytic-MDSCs (M-MDSCs) of naïve, GAS-infected, and GAS-infected mice infused with HSPCs. Bone marrow (BM) populations of (**D**) PMN-MDSCs and (**E**) M-MDSCs of naïve, GAS-infected, and GAS-infected mice infused with HSPCs. (**F**) Heatmap of serum cytokine levels using ClustVis web tool (*Metsalu and Vilo, 2015*). Data representative of two (**A–E**) or four (**F**) independent experiments. (**B–E**) Statistical comparison done using one-way ANOVA with Tukey's correction for multiple comparisons; n = 7 mice per group; *p < 0.05, **p < 0.01. Outliers were identified using the ROUT method (Q = 5%).

The online version of this article includes the following source data and figure supplement(s) for figure 6:

**Source data 1.** BM and PB levels of MDSC populations and cytokine levels raw data.

**Figure supplement 1.** Myeloid-derived suppressor cells (MDSCs) reduce activated T cell proliferation.

**Figure supplement 1—source data 1.** T cell numbers after infusion and suppression assay data.

72 hr after GAS infection showed reduced overall levels of proinflammatory cytokines in GAS-infected mice that received HSPC infusion (*Figure 6F*). These findings suggest that HSPC infusion supports the production of MDSC populations sufficient to dampen maladaptive proinflammatory cues during sepsis.

## Infused HSPCs do not engraft but produce myeloid cells including MDSCs

In order to determine whether HSPCs infused in an infected mouse directly differentiate into MDSCs, we performed lineage tracing experiments using expression of the CD45.1 variant to distinguish infused HSPCs from endogenous CD45.2 cells. Mice were infected with GAS and then rescued with HSPCs 24 hr after GAS inoculation. Thirty days after infection, mice still showed signs of inflammation on the leg, indicating the inflammatory cues that drive HSPC activation and differentiation were still present. At 30 days after infection, the CD45.1+ cell compartment showed no HSCs but a low number of MPP1 and myeloid-biased MPP3s in the BM (*Figure 7A*). Lineage analysis in the BM showed that these cells gave rise to more myeloid cells compared to lymphoid. Furthermore, a fraction of the

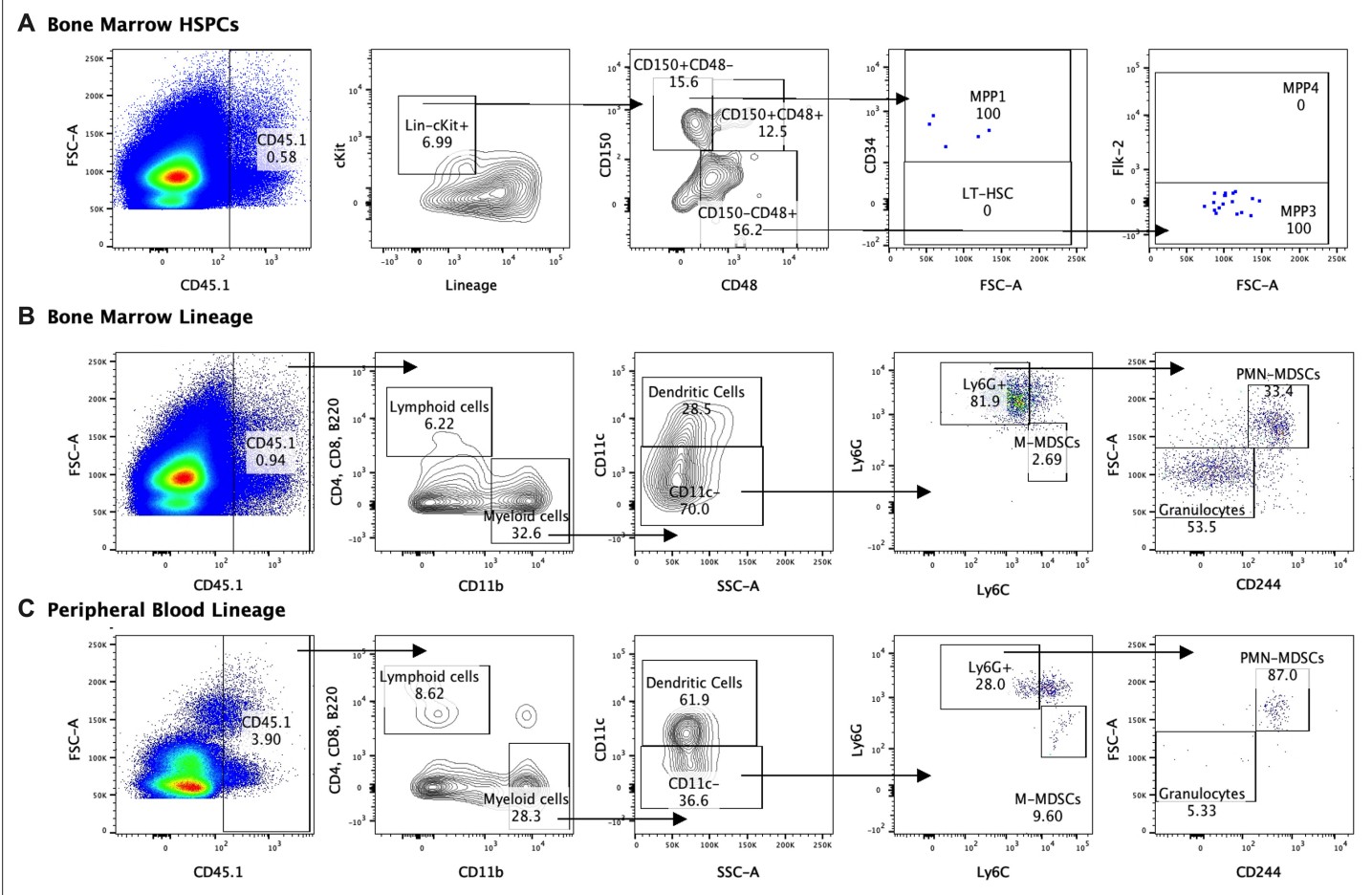

**Figure 7.** Lineage fate of infused hematopoietic stem and progenitor cells (HSPCs) in Group A *Streptococcus* (GAS)-infected mice skews myeloid without signs of stem cell engraftment. Gating representation of the lineage fate of (**A**) bone marrow (BM) HSPCs, (**B**) BM lineage cells, and (**C**) peripheral blood (PB) lineage cells 30 days after GAS infection. Data is representative of three independent experiments.

cells became new monocytic-MDSCs (M-MDSCs) and polymorphonuclear-MDSCs (PMN-MDSCs) (*Figure 7B*). Upon examination of the PB, circulating CD45.1 were primarily myeloid cells, with a small fraction identified as M-MDSCs and PMN-MDSCs (*Figure 7C*). Collectively, these data show that infused cells differentiate toward the myeloid lineage with no sign of stem cell engraftment. Whereas MDSCs did arise directly from infused cells, their numbers were not sufficient to account for the large increase in MDSCs observed in the HSPC-rescued mice. These data suggest that HSPC infusion contributes to MDSC expansion via both direct and indirect mechanisms.

## Discussion

Here, we show HSPC infusion holds therapeutic potential for bacterial sepsis. Our studies demonstrate that GAS infection induces a robust myeloid response just 24 hr after infection and significantly depletes HSPC populations in the BM. After infection, endogenous HSPCs are driven to differentiate toward the myeloid lineage. However, this response is insufficient to prevent disease progression and pathogen dissemination, resulting in mortality in 5–7 days. While sepsis has been described to cause mobilization of HSPCs (*Skirecki et al., 2019*), we did not find any evidence of HSC or MPP mobilization to the spleen. Strikingly, we found infusing just 10,000 HSPCs improved survival in GAS-infected mice and mice with GAS and influenza superinfection. This infusion was capable of increasing PB and BM hematopoietic populations of infected mice. Specifically, HSPC infusion restored BM HSPCs and both PB and BM MDSC populations. Importantly, HSPC infusion did not reduce pathogen burden,

but contributed to survival via the generation of immunoregulatory cells that dampened maladaptive inflammatory signaling in infected mice.

The hematopoietic and immune systems are comprised of immune cells with antimicrobial killing capacity as well as various types of immunomodulatory cells such as MDSCs, regulatory B cells (Bregs), and regulatory T cells (Tregs) (*Maizels and Smith, 2011*; *Rosser and Mauri, 2015*; *Schrijver et al., 2019*). In the short time frame of acute sepsis, myeloid cells such as neutrophils, monocytes, and macrophages are of critical importance in rapidly recognizing and killing invading bacteria. Our data demonstrate that these cells and even the progenitors that produce them in the BM can be rapidly depleted during a severe acute infection. Furthermore, lineage tracing experiments provide the first direct evidence that terminally differentiated myeloid cells are rapidly produced from the level of the HSC during an acute infection. Initially we hypothesized that replacement of HSPCs may improve outcomes from acute bacterial infection by boosting the availability of myeloid cells to kill bacteria. While myeloid cell populations were somewhat restored after HSPC infusion, this was insufficient to reduce pathogen burden.

Dysregulated inflammation is one of the main drivers of morbidity and mortality during infections (*Huang et al., 2005*; *Frank and Paust, 2020*; *Karki et al., 2021*; *Fajgenbaum and June, 2020*). For example, excessive inflammatory responses are a common result of seasonal influenza (*Yu et al., 2011*; *Frank and Paust, 2020*) and SARS-CoV-2 infection (*Karki et al., 2021*). Seasonal influenza increases the susceptibility of patients to secondary bacterial infections or superinfection (*Rynda-Apple et al., 2015*). Superinfections exacerbate the proinflammatory environment of common viral infections and are associated with increased morbidity and mortality (*Rynda-Apple et al., 2015*; *Paget and Trottein, 2019*). To our surprise, HSPC infusion was protective in a model of influenza and GAS superinfection, suggesting that its protective effects are robust even in the setting of severe inflammation. In our mouse models, infection dramatically increased cytokine levels within just 3 days of infection. Interestingly, HSPC infusion was accompanied by an overall decrease in serum cytokine levels and a specific dampening of cytokines involved in 'cytokine storm' (*Huang et al., 2005*; *Karki et al., 2021*; *Fajgenbaum and June, 2020*). HSPCs have been described to produce cytokines, indicating that they have the capacity to direct immune function (*Chen et al., 2016*). However, whether the cytokines produced by HSPCs themselves contribute to the regulation of the immune response has heretofore been unknown. Our data point toward an immunomodulatory role of HSPC infusion that could prevent immune dysregulation during sepsis.

Immunoregulatory cells in the hematopoietic system include Tregs, Bregs, and MDSCs (*Maizels and Smith, 2011*; *Schrijver et al., 2019*; *Uhel et al., 2017*). Perhaps the most recognized immunomodulatory cell known is the Treg. While Tregs have essential roles regulating immune responses to pathogens (*Maizels and Smith, 2011*), we did not see differences in any lymphocyte population, including T cells, that would suggest a Treg-mediated anti-inflammatory mechanism after HSPC infusion. MDSCs are immature myeloid cells that have strong anti-inflammatory roles by suppressing the responses of T-helper cells that contribute to the development of sepsis (*Schrijver et al., 2019*; *Köstlin et al., 2017*; *Delano et al., 2007*). PMN-MDSCs and M-MDSCs have strong anti-inflammatory functions that can be beneficial or detrimental depending on the setting. In fact, some studies have shown that MDSCs contribute to clinical worsening during sepsis (*Schrijver et al., 2019*). For almost three decades, increased circulating immature myeloid cells have been a clinical marker of SIRS (*Bone et al., 1992*). Interestingly, increased MDSCs during sepsis have also been associated with increased development of nosocomial infections (*Uhel et al., 2017*). However, in our GAS model of accelerated infection, the increase in MDSC populations after HSPC infusion was accompanied by lower overall cytokine levels and increased survival, suggesting that the immunomodulatory functions of MDSCs are beneficial during the early stages of systemic inflammation and could prevent sepsis-related mortality (*Chang et al., 2018*).

An important limitation of our study is that the lineage fate and the tissue or organ destination of the infused HSPCs at the early stages of infusion remain unknown. The small number of cells infused makes it challenging to identify them in the pool of endogenous cells of the recipient mice. While our data suggest that infused HSPCs directly and indirectly boost MDSC production by endogenous cells, further work will be required to determine the mechanisms by which HSPCs contribute to MDSC expansion. In addition, further analysis of HSPC subpopulations will be required to determine if long-term HSCs or short-lived MPPs confer the greatest therapeutic potential. Identifying a short-lived

hematopoietic progenitor that can signal endogenous cells to restore MDSC populations could represent a promising alternative therapeutic avenue (*Hidalgo et al., 2019*; *Karpova et al., 2017*; *Nagai et al., 2006*) to treat sepsis while avoiding concern of possible graft versus host disease complications (*Batsali et al., 2020*; *Tijaro-Ovalle et al., 2019*).

Currently, G-CSF, GM-CSF, and granulocyte transfusion (*Robinson and Marks, 2004*; *Price et al., 2015*; *Klein and Castillo, 2017*) are used to prevent or treat sepsis in oncology patients with chemotherapy-induced fever and neutropenia. However, the clinical efficacy of granulocyte transfusion is poor (*Price et al., 2015*; *Klein and Castillo, 2017*). Here, we have shown infusing HSPCs is a promising alternative to granulocyte transfusion. Current granulocyte doses in humans are around $1 \times 10^{10}$ cells per m$^2$ body surface area given daily or every other day (*Price et al., 2015*; *Teofili et al., 2016*). Our infusion model only uses a single dose of $1.7 \times 10^7$ cells per m$^2$ body surface area (or 10,000 HSPCs in a mouse). It is important to emphasize that 10,000 HSPCs is a relatively small number of cells to infuse into a mouse as it represents less than 0.01% of the nucleated BM cells in a mouse. Collectively, the single low HSPC dose compared to multiple larger granulocyte transfusions suggests that HSPCs are more effective than granulocytes, cell for cell, in the treatment of sepsis. While the path to a clinical application can be long, our findings could lead to the future development of a new therapeutic approach that could succeed where granulocyte infusions have fallen short.

# Materials and methods

## Key resources table

| Reagent type (species) or resource | Designation | Source or reference | Identifiers | Additional information |
|---|---|---|---|---|
| Antibody | American Hamster monoclonal Anti-mouse CD3e | Biolegend | Clone 145–2C11 (Cat. No. 100301) | (2 µg/mL) RRID:AB_312666 |
| Antibody | Syrian Hamster monoclonal anti-mouse CD28 | Biolegend | Clone 37.51 (Cat. No. 102101) | (2 µg/mL) RRID:AB_312866 |
| Antibody | Rat Monoclonal anti-mouse Gr1-PECy5 | eBioscience | Clone RB6-8C5 (Cat. No. 15-5931-82) | (1:100) RRID:AB_468813 |
| Antibody | Rat Monoclonal Anti-Mouse CD11b PE-Cyanine5 | eBioscience | Clone M1/70 (Cat. No. 15-0112-82) | (1:100) RRID:AB_468714 |
| Antibody | Rat Monoclonal Anti-Human/Mouse CD45R (B220) PE-Cyanine5 | eBioscience | Clone RA3-6B2 (Cat. No. 15-0452-82) | (1:100) RRID:AB_468755 |
| Antibody | Rat Monoclonal ANTI-MOUSE CD4 PE-Cy5 | eBioscience | Clone GK1.5 (Cat. No. 15-0041-82) | (1:100) RRID:AB_469532 |
| Antibody | Rat Monoclonal Anti-Mouse CD8a PE-Cyanine5 | eBioscience | Clone 53–6.7 (Cat. No. 15-0081-82) | (1:100) RRID:AB_468706 |
| Antibody | Rat Monoclonal Anti-Mouse TER-119 PE-Cyanine5 | eBioscience | Clone TER119 (Cat. No. 15-5921-82) | (1:100) RRID:AB_468810 |
| Antibody | Rat monoclonal anti-mouse Sca-1 Pacific Blue | Biolegend | Clone D7 (Cat. No. 108120) | (1:100) RRID:AB_493273 |
| Antibody | Rat monoclonal anti-mouse Ly-6G | Biolegend | Clone 1A8 (Cat. No. 127605) | (1:100) RRID:AB_1236488 |
| Antibody | Mouse monoclonal Anti-Mouse CD45.1 PE | Biolegend | Clone A20 (Cat. No.110707) | (1:100) RRID:AB_313496 |
| Antibody | Rat Monoclonal anti-mouse CD117 (c-Kit) APC-eFluor 780 | eBioscience | Clone 2B8 (CaT. No. 47-1171-82) | (1:100) RRID:AB_1272177 |

*Continued on next page*

*Continued*

| Reagent type (species) or resource | Designation | Source or reference | Identifiers | Additional information |
|---|---|---|---|---|
| Antibody | Rat Monoclonal Anti-Mouse CD150 PECy7 | Biolegend | Clone TC15-12F12.2 (Cat. No. 115914) | (1:100) RRID:AB_439797 |
| Antibody | American Hamster Monoclonal Anti-mouse CD48 APC | eBioscience | Clone HM48-1 (Cat. No. 17-0481-82) | (1:100) RRID:AB_469408 |
| Antibody | Rat Monoclonal Anti-Mouse CD34 FITC | eBioscience | Clone RAM34 (11-0341-82) | (1:50) RRID:AB_465021 |
| Antibody | Rat Monoclonal Anti-Mouse CD135 DyLight680 | Novus | Clone A2F10 (Cat. No. NBP1-43352FR) | (1:50) RRID:AB_2904163 |
| Antibody | Rat Monoclonal Anti-Mouse CD16/32 BV605 | BD | Clone 2.4G2 (93) (Cat. No. 563006) | (1:100) RRID:AB_2737947 |
| Antibody | Rat Monoclonal Anti Mouse CD41 BV480 | BD | Clone MWReg30 (Cat. No. 746554) | (1:100) RRID:AB_2743844 |
| Antibody | Rat Monoclonal Anti-mouse Ly-6C APC | Biolegend | Clone HK1.4 (Cat. No. 128016) | (1:100) RRID:AB_1732076 |
| Antibody | Rat Monoclonal Anti-mouse CD244 PECy7 | eBioscience | Clone eBio244F4 (Cat. No. 25-2441-82) | (1:100) RRID:AB_2573432 |
| Antibody | Rat Monoclonal Anti-Mouse F4/80 Pacific Blue | ThermoFisher | Clone BM8 (Cat. No. MF48028) | (1:100) RRID:AB_1500083 |
| Chemical compound, drug | Penicillin-Streptomycin | Invitrogen | (Cat. No. 15140122) | |
| Chemical compound, drug | Hanks' Balanced Salt Solution | Gibco/ Thermofisher | HBSS (Cat. No. 14170161) | |
| Chemical compound, drug | HEPES (1M) | Gibco/ Thermofisher | Cat. No. 15630080 | |
| Chemical compound, drug | Tamoxifen | Sigma | (Cat. No. 10540-29-1) | |
| Commercial assay or kit | CD117 Microbeads | Miltenyi-Biotec | (Cat. No. 130-091-224) | |
| Commercial assay or kit | CD3e Microbead Kit, Mouse | Miltenyi-Biotec | (Cat. No. 130-094-973) | |
| Commercial assay or kit | Invitrogen SuperScript IV First-Strand Synthesis System | Invitrogen | (Cat. No. 18091050) | |
| Commercial assay or kit | iTaq Universal SYBR Green Supermix; BioRad | Bio Rad | (Cat. No. 172–5121) | |
| Peptide, recombinant protein | Recombinant Mouse IL-2 | Biolegend | (Cat. No. 575404) | |
| Sequence-based reagent | Nucleoprotein (NP) Forward primer | IDT | NP F1 | 5'-GGGTGAGAATGGACGAAAAAC-3' |
| Sequence-based reagent | Nucleoprotein (NP) Reverse primer | IDT | NP R1 | 5'-GATCCATCATTGCTTTTTGTGCA-3' |
| Software, algorithm | ClustVis | *Metsalu and Vilo, 2015* | ClustVis | RRID:SCR_017133 |
| Strain, strain background (*Mus musculus*) | KRT18-CreERT2: Rosa26-lox-STOP-lox-TdTomato | This paper | | Tamoxifen-inducible Cre system |
| Strain, strain background (*Mus musculus*) | C57Bl/6J | The Jackson Laboratory | Strain #000664 | Wildtype mouse line – CD45.2 RRID:IMSR_JAX:000664 |
| Strain, strain background (*Mus musculus*) | B6.SJL-*Ptprc^a Pepc^b*/BoyJ | The Jackson Laboratory | Strain #002014 | C57Bl/6J congenic strain - CD45.1 RRID:IMSR_JAX:002014 |

*Continued on next page*

*Continued*

| Reagent type (species) or resource | Designation | Source or reference | Identifiers | Additional information |
|---|---|---|---|---|
| Strain, strain background (*Streptococcus pyogenes*) | Group A *Streptococcus* | clinical isolate | MGAS315, *emm3* genotype, Serotype M3 | |
| Strain, strain background (*Orthomyxoviridae, influenza virus A*) | Influenza A virus | ATCC | PR8 H1N1 | |
| Commercial assay or kit | CellTrace Violet Cell Proliferation Kit, for flow cytometry | Invitrogen | (Cat. No. C34571) | |

## Mice

We used WT C57Bl/6 (CD45.2) (RRID:IMSR_JAX:000664) and C57Bl/6.SJL (CD45.1) (RRID:IMSR_JAX:002014) mice 8–10 weeks of age. Lineage tracing using KRT18-CreERT2: Rosa26-lox-STOP-lox-TdTomato we made by crossing KRT18-CreERT2 mice obtained from Dr Daisuke Nakada (Baylor College of Medicine) and Rosa26-lox-STOP-lox-TdTomato mice (stock # 007914) obtained from Jackson Laboratories (Bar Harbor, ME, https://www.jax.org). All mice genotypes were confirmed by polymerase chain reaction (PCR) prior to the start of the experiments. Mice were assigned to each experimental group at random. Both male and female mice were used for all the experiments except for the superinfection survival studies as it has been shown that female mice have long-lasting hyper-responsiveness to respiratory infections (*Larcombe et al., 2011*). Therefore, only male mice were used in the superinfection experiments. Individual mice were assigned to groups randomly and were age and sex-matched for each independent experiment.

## Pathogen inoculation and quantification

Mice were infected with *S. pyogenes* strain MGAS315 by intramuscular injection on the hind limb with $2 \times 10^6$ CFU. To determine the bacterial load, limb, spleen, and blood were collected from infected mice. Limb and spleen tissue were homogenized, serially diluted, and plated on blood agar plates (BAP) (BD, Franklyn Lake, NJ, https://www.bd.com). Blood was serially diluted and plated on BAP. Limb and spleen bacterial load was normalized to the grams of tissue that was homogenized.

Influenza A H1N1 PR8 strain infections were done by intranasal inoculation with 150 PFU. Viral load was quantified by collecting viral particles from lung lavage fluid using Amicon Ultra 0.5 mL (Millipore Sigma, Burlington, MA, https://www.emdmillipore.com), and RNA was purified using the TRIZOL method followed by the quantification of viral particles by real-time PCR of virus-specific nucleoprotein gene. The exact quantity was calculated using a standard curve of purified viral particles with known concentration and normalized by the amount of lung tissue collected.

## HSPC isolation and purification

Six bones were collected from naïve donor mice (two tibias, two femurs, and two hip bones). Bones were then carefully crushed in HBSS media with 1% penicillin/streptomycin and HEPES. Filtered BM was RBC lysed using 5 min RBC lysis buffer (Biolegend, San Diego, CA, https://www.biolegend.com). Lysis buffer was washed out and cells were stained with anti-CD117 magnetic beads (eBiosciences, San Diego, CA, https://www.thermofischer.com) using the manufacturer's protocol. CD117+ cells were positively selected using the AutoMACS instrument (Miltenyi). CD117+ cells were washed and stained for Lineage markers (*Table 1*) and Sca-1. After staining, HSPCs were purified by cell sorting.

## Flow cytometry and cell sorting

Flow cytometry analyses were done using LSR II and BD Fortessa instruments. Cells were identified by the differential expression of markers listed in *Table 1*. Our cocktail of Lineage (Lin) markers include Gr1, CD11b, B220, CD4, CD8, and Ter119.

Cell sorting of HSPCs and their subpopulations were done using the SONY SH800 sorter and the BD FACS Aria Fusion using the markers listed in *Table 1*. Post-sort purity test showed that sorted cells were 95–98% pure.

## Cre induction

Cre activation in KRT18-CreERT2: Rosa26-lox-STOP-lox-TdTomato was induced with tamoxifen. Each mouse was administered tamoxifen by intraperitoneal injection at a dose of 100 mg/kg body weight for 5 consecutive days prior to the start of the lineage tracing experiments.

## Cell infusion

All infusions were done intravenously by retroorbital injection. Rescued mice received cells resuspended in saline solution while the control mice were injected with saline solution alone.

## Cytokine profiling

Serum was collected using a BD Microtainer blood collection tube (San Jose, CA, https://www.bdbio-sciences.com). Serum levels of cytokines were analyzed through Eve Technologies company (Calgary, AB, Canada, https://www.evetechnologies.com).

## T cell suppression assay

T cells were isolated from the spleen using anti-CD3 magnetic beads from Miltenyi Biotec (Bergisch Gladbach, Germany, https://www.miltenyibiotec.com) and MDSCs were sorted using the SONY SH800 sorter and the BD FACS Aria Fusion as described above. T cell were activated with anti-CD3 and anti-CD28 coated plates and supplemented with IL-2 to support proliferation and then cultured alone of with M-MDSCs or PMN-MDSCs. T cells were stained with CellTrace Violet and proliferation was measured by dye dilution using flow cytometry.

## Statistical tests

Normality was assessed using the Shapiro-Wilk test and variances were compared using F-tests. Comparisons between two groups were made done using unpaired $t$-test for parametric data, Welch's $t$-test for parametric data without equal variances, and Mann-Whitney test for non-parametric data. Tests involving three or more comparisons were done using one-way ANOVA with Tukey's correction for multiple comparisons or Kruskal-Wallis test with Dunn's correction for multiple comparisons. Comparisons of survival curves were done using Mantel-Cox tests. Outliers were identified using the ROUT method (Q = 5%). Graphs are shown as mean ± SEM. Sample size of each experiment was calculated based on pilot experiments and using an alpha = 0.05 and power = 0.80. Each specific statistical test used as well as group size and independent experiments are described on each figure legend.

## Study approval

Mice are housed in AAALAC-accredited, specific-pathogen-free animal facilities at Baylor College of Medicine and Texas Children's Hospital. All experiments are approved and follow the guidelines stated in our protocol approved by the Institutional Animal Care and Use Committee (IACUC) and by the Baylor College of Medicine institutional review board.

# Acknowledgements

The authors would like to thank Catherine Gillespie, Maksim Mamonkin, Meghan Kisiel, Olumide Ayeni, and members of the King lab for their input into the study. DEMM and KYK were supported by the NIH grants R01HL136333, R01HL134880, and R35HL155672 (KYK). DEMM was also supported by the Howard Hughes Medical Institute (HHMI) Gilliam Fellowship for Advanced Study. This project was supported by the Cytometry and Cell Sorting Core at Baylor College of Medicine with funding from the CPRIT Core Facility Support Award (CPRIT-RP180672), the NIH (CA125123 and RR024574), and the assistance of Joel M Sederstrom. This project was also assisted by the Dan L Duncan Cancer Center and the William T Shearer Center for Human Immunobiology.

## Additional information

### Funding

| Funder | Grant reference number | Author |
| --- | --- | --- |
| National Institutes of Health | R01HL136333 | Katherine Y King |
| National Institutes of Health | R01HL134880 | Katherine Y King |
| National Institutes of Health | R35HL155672 | Katherine Y King |
| Howard Hughes Medical Institute | Gilliam Fellowship | Daniel E Morales-Mantilla |
| Cancer Prevention and Research Institute of Texas | CPRIT-RP180672 | Katherine Y King |
| National Institutes of Health | CA125123 | Katherine Y King |
| National Institutes of Health | RR024574 | Katherine Y King |

The funders had no role in study design, data collection and interpretation, or the decision to submit the work for publication.

### Author contributions

Daniel E Morales-Mantilla, Conceptualization, Data curation, Formal analysis, Investigation, Methodology, Visualization, Writing - original draft, Writing - review and editing; Bailee Kain, Formal analysis, Investigation, Methodology; Duy Le, Investigation, Methodology; Anthony R Flores, Silke Paust, Conceptualization, Resources; Katherine Y King, Conceptualization, Funding acquisition, Project administration, Supervision, Writing - original draft, Writing - review and editing

### Author ORCIDs

Daniel E Morales-Mantilla (ID) http://orcid.org/0000-0002-3459-6231
Katherine Y King (ID) http://orcid.org/0000-0002-5093-6005

### Ethics

This study was performed in strict accordance with the recommendations in the Guide for the Care and Use of Laboratory Animals of the National Institutes of Health. All of the animals were handled according to approved institutional animal care and use committee (IACUC) protocol (#4802) of Baylor College of Medicine.

### Decision letter and Author response

Decision letter https://doi.org/10.7554/eLife.74561.sa1
Author response https://doi.org/10.7554/eLife.74561.sa2

## Additional files

### Supplementary files

• Transparent reporting form

### Data availability

All data generated or analysed during this study are included in the manuscript and supporting file.

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
