## [Editor Report]

This preclinical study reports on a novel strategy for sepsis. Sepsis induced by Group A Streptococcus (GAS) in mice leads to depletion of bone marrow HSPCs and mortality and infusion of naive donor HSPCs lower mortality but has no effect on bacterial burden. This supports that HSPCs infusion might attenuate the detrimental immune response in sepsis warranting further investigation of this novel concept.

---

## [Decision Letter]

**Decision letter after peer review:**

Thank you for submitting your article "Hematopoietic stem and progenitor cells improve survival from sepsis by boosting immunomodulatory cells" for consideration by *eLife*. Your article has been reviewed by 2 peer reviewers, and the evaluation has been overseen by a Reviewing Editor and Jos Van der Meer as the Senior Editor. The reviewers have opted to remain anonymous.

Essential revisions:

General comments:

The reviewers comment that the authors are a bit too optimistic in stating "Our findings could lead to the development of an efficacious new therapeutic approach that could succeed where granulocyte infusions have fallen short". There are so many steps and hurdles that need to be taken before this kind of intervention could be translated to the clinic.

In addition, the data presented in figure 7 is of little value, as these were obtained in non-inflamed/infected mice. Therefore, conclusions such as "Whereas MDSCs did arise directly from infused cells, their numbers were not sufficient to account for the large increase in MDSCs observed in the HSPC-rescued mice. These data suggest that HSPC infusion contributes to MDSC expansion via both direct and indirect mechanisms." are not justified.

Please adjust this conclusion and soften the statements regarding the translation to the clinical setting throughout the manuscript.

More specific comments that need to be addressed:

1) Can the authors explain why MPP4 is also depleted following GAS (Figure 2F), as this is a lymphoid progenitor rather than a myeloid one. This appears not to be congruent with the lack of depletion of BM T- and B-cells (Figure 1E-F).

2) In Figure 2, why were different gating/markers for MPP2/3/4 used in the BM compared with the spleen?

3) The authors state: "This finding suggests that the loss of BM HSPC populations is not simply a result of migration from the BM into the spleen" However, there is a significant increase in HSPCs in the spleen (Figure 2G), so why discard this possibility? How can these number/proportions be directly compared?

4) Spleen subpopulations in Figure 2 are expressed as % of total spleen cells, but there is no data showing whether or not there was there a change in total spleen cells?

5) What is shown in figure 4B? Which organ?

6) At what timepoint were samples taken for the analyses presented in Figure 6A-E?

7) The authors need to explain why they used the specific model of sepsis with *Streptococcus pyogenes* instead of the classical models of LPS injection or whole bacteria injection or cecal ligation and puncture. They also need to provide evidence that the studied model was accompanied by organ failure through the measurement of dry to wet lung ratio, serum creatinine and serum aminotransferases.

8) A detailed methodology on the isolation of progenitor cells is needed. The authors need also to explain if they used an autograph or allograph approach.

9) How did the authors adjust for any graft versus host reaction?

10) Where other bacterial species isolated from the liver? Why were not kidney and lung cultures being performed?

---

## [Author Response]

Essential revisions:General comments:The reviewers comment that the authors are a bit too optimistic in stating "Our findings could lead to the development of an efficacious new therapeutic approach that could succeed where granulocyte infusions have fallen short". There are so many steps and hurdles that need to be taken before this kind of intervention could be translated to the clinic.

We appreciate the help in making this part of the conclusions clearer. We are aware of the multitude of steps needed before a clinical application and we have rephrased our conclusion as:

“While the path to a clinical application can be long, our findings could lead to the future development of a new therapeutic approach that could succeed where granulocyte infusions have fallen short.”

In addition, the data presented in figure 7 is of little value, as these were obtained in non-inflamed/infected mice. Therefore, conclusions such as "Whereas MDSCs did arise directly from infused cells, their numbers were not sufficient to account for the large increase in MDSCs observed in the HSPC-rescued mice. These data suggest that HSPC infusion contributes to MDSC expansion via both direct and indirect mechanisms." are not justified.Please adjust this conclusion and soften the statements regarding the translation to the clinical setting throughout the manuscript.

We thank the reviewer for this comment. With regards to figure 7, these mice were in fact infected and showed clear signs of inflammation (swelling, redness, abscess). They were infected at day 0, infused with HSPCs at day 1, and analyzed at day 30 for BM and peripheral blood phenotypes. Even at day 30, the mice had a residual abscess and thus the data reflect an inflamed/infected environment.

In order to clarify this point, we have included a more detailed timeline and treatment description of the mice in the Results section of the manuscript as follows:

“In order to determine whether HSPCs infused in an infected mouse directly differentiate into MDSCs, we performed lineage tracing experiments using expression of the CD45.1 variant to distinguish infused HSPCs from endogenous CD45.2 cells. Mice were infected with GAS and then rescued with HSPCs 24h after GAS inoculation. Thirty days after infection, mice still showed signs of inflammation on the leg, indicating the inflammatory cues that drive HSPC activation and differentiation were still present. At thirty days after infection, the CD45.1+ cell compartment showed no HSCs but a low number of MPP1 and myeloid biased MPP3s in the BM (Figure 7A).”

More specific comments that need to be addressed:1) Can the authors explain why MPP4 is also depleted following GAS (Figure 2F), as this is a lymphoid progenitor rather than a myeloid one. This appears not to be congruent with the lack of depletion of BM T- and B-cells (Figure 1E-F).

We thank the reviewer for this observation. While MPP4s are certainly lymphoid biased, they are known to have plasticity in their differentiation pathways. Furthermore, the lymphoid compartment (T and B cells) tends to be much longer-lived than myeloid cells such as neutrophils and monocytes. In addition, myeloid cells are massively mobilized during an acute infection whereas the shifts in lymphoid cells are more limited. For all of these reasons, an acute depletion of lymphoid progenitors does not necessarily result in an immediate loss of T and B cells.

2) In Figure 2, why were different gating/markers for MPP2/3/4 used in the BM compared with the spleen?

These MPPs are extremely rare in the spleen. Using BM markers would obviate the detection of any cells in the spleen. Thus, we decided to include a broader population in the spleen to detect these populations. Other groups studying spleen hematopoietic cells have also used these markers to describe spleen HSPCs and HSCs (Oda et al., *Scientific Reports*, 2018; Burberry et al., *Cell Host and Microbe*, 2014).

3) The authors state: "This finding suggests that the loss of BM HSPC populations is not simply a result of migration from the BM into the spleen" However, there is a significant increase in HSPCs in the spleen (Figure 2G), so why discard this possibility? How can these number/proportions be directly compared?

We agree with the reviewer that this possibility should not be discarded. However, that magnitude of BM depletion and the small or no changes in the corresponding spleen populations suggests that this is not solely by migration. We appreciate the reviewers pointing this out and have amended the manuscript to read:

“These findings suggest that the loss of BM HSPC populations is not principally a result of migration from the BM into the spleen, and other mechanisms such as terminal differentiation also likely contribute to the noted HSPC depletion, as observed in other studies.”

4) Spleen subpopulations in Figure 2 are expressed as % of total spleen cells, but there is no data showing whether or not there was there a change in total spleen cells?

Absolute numbers in the spleen are not available. However, we did measure total spleen size to determine whether there was a change in cellularity of the spleen.

Author response image 1 shows spleen sizes of naïve and GAS-infected mice 72hrs post-infection. We did not detect any difference in spleen size or weight 24-72 hours after GAS infection, therefore it is reasonable to conclude that a change in % of total spleen cells does reflect a corresponding change in absolute number.

**Author response image 1. sa2fig1:** 

5) What is shown in figure 4B? Which organ?

As stated in the figure legend, the figure shows a “Flow plot of HSPC gating and representation of different surface expressions of cKit and Sca-1 during infection. Plots are gated from lineage negative BM cells.” The organ of origin of these cells is the bone marrow (BM).

6) At what timepoint were samples taken for the analyses presented in Figure 6A-E?

This data was collected 3 days post-infection and 2 days after HSPC infusion. We have stated this more clearly in the manuscript as follows:

“Upon analysis of PB and BM populations 3 days after infection and 2 days after HSPC infusion…”

7) The authors need to explain why they used the specific model of sepsis with *Streptococcus pyogenes* instead of the classical models of LPS injection or whole bacteria injection or cecal ligation and puncture. They also need to provide evidence that the studied model was accompanied by organ failure through the measurement of dry to wet lung ratio, serum creatinine and serum aminotransferases.

We appreciate this observation. We mention in the introduction that GAS is a clinically relevant strain that causes a significant disease burden (Gonzalez-Abad and Sanz, Anales de Pediatría, 2020; Terao and Kawabata, Tanpakushitsu Kakusan Koso, 2009; Beres et al., mBio, 2016; Flores and Shelburne, Clinical Infectious Diseases, 2019). It also induces an IFN-mediated immune response, which, in light of prior studies, led us to hypothesize that HSPCs might have a role to play in the response to GAS (Baldridge et al., Trends in Immunology, 2011; Hormaechea-Agulla et al., Cell Stem Cell, 2021; Matatall et al., Stem Cells, 2014; Matatall et al., Cell Reports, 2016).

We did carefully examine the impact of GAS infection on various organs. While at 3 days post-infection, we did not see a significant difference in serum creatinine and aminotransferases, we did see a significant drop in the albumin-globulin ratio (AGR). Low AGR has been described to be a biomarker of sepsis (Lu et al., *Translational Andrology and Urology*, 2020). Systemic inflammation (Figure 6F) and pathogen dissemination (Figure 4J) are also signs of sepsis.

8) A detailed methodology on the isolation of progenitor cells is needed. The authors need also to explain if they used an autograph or allograph approach.

We have expanded on our strategy for HSPC isolation in the methods section subtitled “*HSPC isolation and purification”* in order to provide a thorough description of the methods. The infused HSPCs are from naïve donor mice, thus, this is an allogeneic infusion.

9) How did the authors adjust for any graft versus host reaction?

The congenic bone marrow transplant model used in our studies are a well-established system developed over 30 years ago which relies on a variant of a blood epitope that is not immunogenic and therefore does not cause GVHD (Brochu S. Blood 1994 84(9):3221-8). In the setting of GVHD, the immune response of the “graft” against the host can be evidenced by increased inflammation and organ failure if left unattended. However, in our model, we see an overall decrease in inflammation and increased survival of mice for over a month, which also supports the well-established concept that there is no immune response from the infused cells against host cells.

10) Where other bacterial species isolated from the liver? Why were not kidney and lung cultures being performed?

We cultured limb tissue to determine bacterial load as this is the initial site of infection. We cultured the spleen as this is the most likely site of pathogen dissemination during a systemic infection. Spleen data suggests that there is in fact dissemination of GAS to other organs. Note that bacterial culture on nonselective media did not show other bacterial species in the spleen, therefore we would not suspect other bacterial species in the liver. Furthermore, histology did not show other bacterial species in the liver.